# The Potential Role of SARS-CoV-2 Infection and Vaccines in Multiple Sclerosis Onset and Reactivation: A Case Series and Literature Review

**DOI:** 10.3390/v15071569

**Published:** 2023-07-18

**Authors:** Eleonora Tavazzi, Anna Pichiecchio, Elena Colombo, Eleonora Rigoni, Carlo Asteggiano, Elisa Vegezzi, Francesco Masi, Giacomo Greco, Stefano Bastianello, Roberto Bergamaschi

**Affiliations:** 1IRCCS Mondino Foundation, 27100 Pavia, Italy; anna.pichiecchio@mondino.it (A.P.); elena.colombo@mondino.it (E.C.); eleonora.rigoni@mondino.it (E.R.); carlo.asteggiano@mondino.it (C.A.); elisa.vegezzi@mondino.it (E.V.); francesco.masi01@universitadipavia.it (F.M.); giacomo.greco01@universitadipavia.it (G.G.); roberto.bergamaschi@mondino.it (R.B.); 2Department of Brain and Behavioural Sciences, University of Pavia, 27100 Pavia, Italy; stefano.bastianello@unipv.it

**Keywords:** multiple sclerosis, myelitis, CNS dysimmune disorders, COVID-19, SARS-CoV-2, vaccines

## Abstract

The recent SARS-CoV-2 pandemic and related vaccines have raised several issues. Among them, the potential role of the viral infection (COVID-19) or anti-SARS-CoV-2 vaccines as causal factors of dysimmune CNS disorders, as well as the safety and efficacy of vaccines in patients affected by such diseases and on immune-active treatments have been analyzed. The aim is to better understand the relationship between SARS-CoV-2 infection/vaccines with dysimmune CNS diseases by describing 12 cases of multiple sclerosis/myelitis onset or reactivation after exposure to SARS-CoV-2 infection/vaccines and reviewing all published case reports or case series in which MS onset or reactivation was temporally associated with either COVID-19 (8 case reports, 3 case series) or anti-SARS-CoV-2 vaccines (13 case reports, 6 case series). All the cases share a temporal association between viral/vaccine exposure and symptoms onset. This finding, together with direct or immune-based mechanisms described both during COVID-19 and MS, claims in favor of a role for SARS-CoV-2 infection/vaccines in unmasking dysimmune CNS disorders. The most common clinical presentations involve the optic nerve, brainstem and spinal cord. The preferential tropism of the virus together with the presence of some host-related genetic/immune factors might predispose to the involvement of specific CNS districts.

## 1. Introduction

The advent of the SARS-CoV-2 pandemic has raised multiple issues, and among them, some relevant ones on the relationship between the virus and dysimmune diseases. In particular, considering the profound alteration of the immune system induced by the virus [1] as well as its neurotropism, confirmed by the presence of neurological symptoms in the acute and post-acute phase of SARS-CoV-2-related infection (COVID-19) [2], the center of a lively debate has been the possible bidirectional relationship between the virus and central nervous system (CNS) disorders [3,4,5]. More specifically, researchers have been investigating whether SARS-CoV-2 is able to cause or contribute to the onset of dysimmune central nervous system (CNS) disorders, as well as whether people already affected with such diseases and likely to be on immune-active treatments were exposed to a higher risk of severe infection [6,7].

The possible association between viral infections and the onset of autoimmunity in different organs is well known and has been extensively studied [8,9,10]. In particular, with respect to multiple sclerosis (MS), recent studies reported a close relationship between Epstein–Barr virus (EBV) infection and MS prevalence both in experimental and in population-based studies [11,12,13,14]. This finding reinforced the hypothesis of the role of a viral agent as the trigger of a dysimmune, self-sustaining process eventually leading to an inflammatory demyelinating CNS disorder. The advent of COVID-19 and the occurrence of demyelinating inflammatory disorders temporally following the acute infection have further confirmed the potential role of viruses as triggers of neuroinflammation [12,15,16,17].

The subsequent advent of vaccines against SARS-CoV-2 had a twofold consequence: On one hand, it provided an effective strategy against the virus, dramatically reducing the incidence of SARS-CoV-2-related infection (COVID-19) [18,19,20,21]. On the other hand, though, it raised several safety concerns: Could the vaccine trigger a dysimmune response? Could it worsen the disease course of pre-existing dysimmune conditions? Another relevant issue regarded the ability of people already treated with immunomodulating or immunosuppressive drugs to develop an effective immune response capable of counteracting the viral infection. Research on these aspects has contributed to clarifying several aspects, such as the SARS-CoV-2 entry route into the CNS, the potential mechanisms implicated in the SARS-CoV-2-related autoimmune response, as well as the immune system components activated by different vaccine modalities [22].

With respect to MS, the literature data concordantly reported a similar incidence and mortality rate related to COVID-19 in people with MS (pwMS) with respect to the general population [23,24,25], identifying older age, greater disability, presence of comorbidities and anti-CD20 therapies as risk factors for a more severe infection and a higher risk of hospitalization [26,27,28]. Analogously, the humoral and cellular responses to vaccines have been investigated in pwMS with encouraging findings of effective vaccinal protection, with the exception of pwMS on anti-CD20 therapies, which are therapies specifically targeting BN lymphocytes [29]. As expected, these patients have markedly reduced antibody production, but vaccination does produce a preserved cellular-mediated immunological response [27,30,31,32,33,34].

Despite the considerable effort invested during the recent pandemic, several issues remain at least partially unsolved, especially regarding the potential pathogenetic role of SARS-CoV-2 in MS onset or worsening, together with safety concerns with respect to vaccines in pwMS [35]. These aspects are of great importance as vaccine hesitancy continues to remain a challenge, even nowadays [36,37].

We here report a case series of subjects that presented with a first clinical manifestation compatible with MS or an MS/myelitis relapse after SARS-CoV-2 infection or vaccine exposure. The aim of describing these clinical cases is not to argue in favor of an increased risk of post-infectious/post-vaccinal dysimmune events but rather to shed further light on the possible pathogenetic mechanisms of CNS demyelinating disorders associated with SARS-CoV-2 and the related vaccines.

## 2. MS Onset after SARS-CoV-2 Exposure

Case 1: A previously healthy unvaccinated 38-year-old man presented with diplopia, 14 days after COVID-19 infection, diagnosed through a nasopharyngeal swab for a SARS-CoV-2 reverse-transcriptase-polymerase-chain-reaction assay. Brain MRI performed three weeks later revealed multiple T2-hyperintense and T1-hypointense lesions, primarily located in the supratentorial region, showing predominant periventricular distribution, and at least one concomitant infratentorial lesion in the posterior portion of the pons (Figure 1). Spine MRI was normal. CSF examination revealed 15 cells (lymphocytes) and the presence of oligoclonal bands; an extensive viral assessment was negative. Serum anti-MOG and anti-AQP4 antibodies were negative. The patient was treated with i.v. high-dose steroids (6-methylprednisolone (MP) 1 gr/daily for 3 days) with a gradual and complete recovery (Expanded Disability Status Scale (EDSS) 1.0 for brisk reflexes). The patient was diagnosed with MS and started dimethyl fumarate. At the most recent follow-up examination 20 months after the onset, the patient was completely asymptomatic, with complete stability of brain MRI and neurological examination.

Case 2: A previously healthy unvaccinated 30-year-old man was admitted to our Institute for the acute onset of blurred vision in right eye with retro-orbital pain 1 week after symptomatic COVID-19 (fever, anosmia and ageusia), diagnosed through a nasopharyngeal swab for a SARS-CoV-2 reverse-transcriptase-polymerase-chain-reaction assay. The visual-evoked potential confirmed the diagnosis of optic neuritis in the right eye; neurological examination at admission was otherwise normal. Brain MRI revealed only one T2-hyperintense lesion located in the right cerebellar hemisphere and mild T2-hyperintensity of the right optic nerve. Spine MRI showed a single lesion at the level of C2–C3, with swelling of the spinal cord but without contrast enhancement (CE) (Figure 2A–C). The lumbar puncture showed a mirror pattern, and anti-MOG and anti-AQP4 antibodies were negative. Treatment included i.v. high-dose steroids (6-MP 1 gr/daily for 3 days), with complete recovery. He was diagnosed with MS and started dimethyl fumarate, with no evidence of disease activity 23 months after the beginning of therapy.

Case 3: A 39-year-old unvaccinated woman affected by Hashimoto’s thyroiditis presented reduced visual acuity, dyschromatopsia and pain at ocular movement in the left eye two weeks after a mildly symptomatic COVID-19, diagnosed through a nasopharyngeal swab for a SARS-CoV-2 reverse-transcriptase-polymerase-chain-reaction assay. Ophthalmologic evaluation revealed left eye papilledema and central scotoma. Visual-evoked potentials confirmed the clinical suspicion of optic neuritis. Neurological evaluation showed only brisk reflexes. Brain MRI showed white matter lesions, located in the subcortical and periventricular white matter and in the infratentorial region, with swelling and T2-hyperintensity of the left optic nerve, without CE (Figure 3). CSF analysis detected oligoclonal bands, while anti-AQP4 and anti-MOG autoantibodies were negative.

I.V. high-dose steroids (6-MP 1 gr/daily for 3 days) were started with complete recovery. The patient was diagnosed with relapsing–remitting MS and enrolled in a phase III double-blind clinical trial, randomly assigned to fenebrutinib or teriflunomide treatment.

Case 4: A 29-year-old woman with a previous diagnosis of rheumatoid arthritis treated with low-dose methotrexate presented with sensory hypoesthesia and numbness in both feet rapidly ascending to both limbs and the perineal region which occurred one month after a mildly symptomatic COVID-19 infection, diagnosed through a nasopharyngeal swab for a SARS-CoV-2 reverse-transcriptase-polymerase-chain-reaction assay. Seven months earlier, the patient had completed a full vaccination cycle (BNT162b2 mRNA COVID-19 vaccine, 3 doses), without any problem. A brain MRI showed only a few small T2-hyperintense lesions mainly located in the subcortical white matter, whereas a spine MRI revealed a T2-hyperintense, CE- lesion at the T6 level. Oligoclonal bands were detected in the CSF; serum anti-MOG and anti-AQP4 antibodies were negative. I.V. high-dose steroids (6-MP 1 gr/daily for 5 days) were started with complete recovery. The patient was diagnosed with MS, and the therapeutic strategy agreed to with rheumatologists was to withdraw methotrexate and start cladribine. The patient has been completely asymptomatic since then.

## 3. MS Relapses after COVID-19 Infection

Case 5: A 43-year-old man with a 28-year-long history of MS characterized by clinical and MRI stability and a low level of disability (EDSS 2.0) despite persistently refusing disease-modifying treatments (DMTs) underwent a complete cycle of anti-SARS-CoV-2 mRNA vaccinations (BNT162b2). One month after the third dose, he developed an asymptomatic SARS-CoV-2 infection, diagnosed through a nasopharyngeal swab for a SARS-CoV-2 reverse-transcriptase-polymerase-chain-reaction assay. Three weeks later, he developed marked loss of balance and fatigue, resulting in frequent falls. Once negative to the PCR test, he underwent i.v. high-dose steroids (6-MP 1 gr/daily for 5 days) followed by neuromotor rehabilitation, with only partial recovery (EDSS 4.5) and clinical stability since then. MRI was not performed. The patient has been clinically stable since then.

## 4. MS Onset after Vaccine Exposure

Case 6: A previously healthy 58-year-old man developed gait ataxia, with moderate trunk instability and three episodes of bowel incontinence, two weeks after the second dose of SARS-CoV-2 mRNA vaccine (BNT162b2). Two months later, he underwent a brain MRI showing multiple supra- and infratentorial white matter lesions (Figure 4A). A neurological evaluation performed one month later revealed the presence of pyramidal signs, limb and gait ataxia, with a mild decrease in touch on both feet. Spine MRI documented numerous white matter lesions without contrast enhancement (Figure 4B); CSF analysis detected oligoclonal bands. Autoimmunity screening and serum anti-MOG and anti-AQP4 antibodies were negative. The patient was treated with i.v. steroids (6-MP 1 gr/daily for 5 days) with slight improvement in gait ataxia. He was diagnosed with MS and teriflunomide was started.

Case 7: This clinical case has been the subject of a previous publication to which we refer for further details [38]. Briefly, 4 days after the AZD1222 vaccine, a previously healthy 44-year-old woman developed sensory symptoms starting in her feet and rapidly ascending, to involve the lower back, the perineal region and both lower limbs. Acute transverse myelitis was diagnosed based on clinical symptoms and two single-metamer T2-hyperintense lesions at D7-D8 and D10-D11, with patchy contrast enhancement of the former and, concomitantly, no brain MRI lesions. The patient was treated with i.v. high-dose steroids (6-MP × 5 days) followed by oral tapering with a gradual complete recovery of all the symptoms. At a follow-up brain and spine MRI, one year later, five new lesions were found at a spine level, together with one supratentorial lesion. Repeated CSF analysis revealed the presence of oligoclonal bands. Serum anti-MOG and anti-AQP4 antibodies were negative. The patient was then diagnosed with MS, and natalizumab was started. She has been completely stable from a clinical and radiological point of view since then.

## 5. MS Relapses after Vaccine Exposure

Case 8: A 30-year-old woman, with a history of a completely recovered optic neuritis from 4 years earlier and a brain MRI showing some T2-hyperintense lesions involving the periventricular white matter and the corpus callosum. She had been completely stable since and received two doses of anti-SARS-CoV-2 mRNA vaccine (BNT162b2) in February 2021. Two weeks later, she developed paresthesias of the lower limbs, rapidly extending to the trunk and the upper limbs. Brain MRI showed three new periventricular non-enhancing lesions (Figure 5A), and spine MRI showed one CE-cervical lesion and two dorsal lesions without contrast enhancement (Figure 5B). CSF analysis revealed the presence of oligoclonal bands, while anti-MOG and anti-AQP4 antibodies were negative. She was treated with i.v. high-dose steroids (6-MP × 5 days), with gradual recovery. The patient was then diagnosed with MS and dimethyl fumarate was started, with clinical and radiological stability over time.

Case 9: A 36-year-old man, diagnosed with relapsing–remitting MS in 2012, treated with natalizumab since February 2016, presented with oscillopsia and visual loss in his right eye three days after the first dose of SARS-CoV-2 mRNA vaccine (BNT162b2); he recovered completely after i.v. steroids (6-MP 1 gr/daily for 3 days). Six weeks later, he was administered the second dose of the same vaccine, which was followed, one week later, by the appearance of bilateral feet numbness together with Lhermitte’s sign. Brain and spine MRI revealed multiple lesions mainly located in the periventricular region and a single lesion in the cervical spinal cord at the C2 level, as well as slight T2-hyperintensity without contrast enhancement of the left optic nerve. Multiple lesions in the brain and the cervical lesion showed enhancement after gadolinium injection (Figure 6A,B). The patient was treated with i.v. steroids (6-MP 1 gr/daily for 5 days) with partial recovery of the clinical symptoms. After 5 months of clinical stability, the patient presented with sensory disturbances of the right upper and lower limbs, associated with motor deficit of the upper right limb. He was prescribed a cycle of i.v. steroids (6-MP 1 gr/daily for 6 days) and switched to alemtuzumab, with complete clinical and MRI stability since then.

Case 10: A 40-year-old man with a previous history of inflammatory myelitis having occurred 3 years earlier, characterized by completely recovered mild sensory disturbances and a single monomeric T2-hyperintense lesion located at the C3 level, received his first dose of SARS-CoV-2 mRNA vaccine (BNT162b2). Eleven days later, he presented with numbness in his feet, extending, in the following days, to the lower limbs bilaterally as well as to the perineal region, associated with Lhermitte’s sign. Spine MRI documented the appearance of multiple new lesions in both the cervical and thoracic spinal cord, including at least two enhancing lesions at the level of C2 and of C4-C5 (Figure 7). Neither brain lesions nor oligoclonal bands were detected. The patient was treated with high-dose steroids (6-MP 1 gr/daily for 6 days) followed by oral tapering with complete recovery. The patient started rituximab and has been clinically and radiologically stable since then.

## 6. MS Diagnosis after COVID-19, with Probable Onset after SARS-CoV-2 Vaccines

Case 11: A previously healthy 28-year-old woman presented with sudden onset of numbness and hypoesthesia in her right leg, subsequently extended to the homolateral trunk and upper limb and the contralateral inferior limb, associated with decreased bladder/bowel sensation 5 days after the onset of COVID-19. One month earlier, she complained of a mild numbness in inferior limbs bilaterally; this occurred 2 days after the first dose of SARS-CoV-2 vaccine (BNT162b2), completely and spontaneously recovered, and was not investigated further.

Brain MRI showed several T2-hyperintense lesions located both at the supra- and infratentorial level (brainstem and cerebellum); spine MRI revealed multiple T2-hyperintense lesions located at C3-C4 and C5 and a CE lesion at the C7-T1 level. CSF examination was within normal range, positive for oligoclonal bands. Serum anti-MOG and anti-AQP4 antibodies were negative. The patient was treated with i.v. high-dose steroids (6-MP 1 gr/daily for 5 days) followed by oral steroids tapering, with a mild residual hypoesthesia in the right lower limb together with pyramidal signs, limb. The patient received a diagnosis of MS and natalizumab was initiated.

Case 12: Four days after recovery from a mild COVID-19 infection, a 64-year-old man presented with subacute onset of horizontal binocular diplopia. However, the patient reported two previous clinical events characterized by mild numbness and hypoesthesia in both feet, recovered completely and spontaneously in 2 weeks and was not investigated, both occurring some days after each of the two doses of the SARS-CoV-2 viral-vector vaccine (AZD1222).

The neurological examination showed left internuclear ophthalmoplegia, vertical nystagmus and right pyramidal signs. Brain MRI showed multiple supratentorial and infratentorial T2-hyperintense white matter lesions, mainly located in the periventricular region with multiple CE lesions both in the brain and spinal cord, with a spinal cord lesion burden of three dorsal lesions (Figure 8). CSF analysis was positive for oligoclonal bands; screening for autoimmunity was negative. The patient was treated with i.v. steroids (6-MP 1 gr/daily for 5 days) with complete recovery of the visual disturbance. He was also diagnosed with relapsing–remitting MS and ozanimod was started with complete clinical and MRI stability since then.

## 7. Discussion

Since the beginning of the SARS-CoV-2 pandemic, MS has been listed among the nosological entities requiring careful observation. Several studies have been carried out, attempting to verify whether the incidence of MS and the relapse rate in pwMS were influenced by the quickly spreading viral infection. Data in this regard are contradictory, with some studies finding an increased risk of contrast-enhancing lesions, relapses and disability accrual [39,40,41] as well as an association between COVID-19 severity and MS worsening [42], while some others do not report any impact of the infection on MS disease course [43,44,45]. Analogous contradicting findings have been published with respect to the relationship between vaccines and the risk of MS reactivation [46,47,48,49,50].

Even though it is impossible to attribute a definite causal role for MS onset to SARS-CoV-2 and the related vaccines, several cases have been reported both after COVID-19 (8 case reports, 3 case series, Table 1) and after anti-SARS-CoV-2 vaccines exposure (13 case reports, 6 case series, Table 2) [49,51,52,53,54,55,56,57,58,59,60,61,62,63,64,65,66,67,68,69,70,71,72,73,74,75,76]. Cases of post-viral infection MS onset or relapse presented with clinical symptoms suggestive for demyelinating disorders after a few days to 6 weeks from viral exposure, except for one case in which the viral infection preceded the disease onset by 6 months [58]. The first clinical symptoms suggestive for MS were related to the infratentorial regions (four cases with diplopia, one case with facial palsy and one case with nystagmus and ataxia), the optic nerve (two cases) or the spinal cord (two cases of bilateral sensory symptoms). In the case series published by Avila et al. [59], 20% of pwMS presented with symptoms related to brainstem impairment, 40% with optic neuritis and the remaining 40% with symptoms attributable to the spinal cord. With respect to the two published cases of MS reactivation after viral exposure, one presented with unilateral motor symptoms and one with bilateral sensory symptoms attributable to a documented lesion within the spinal cord.

All the post-SARS-CoV-2 MS onset/reactivations showed CE lesions together with non-CE lesions. This finding satisfies the criterion of dissemination in the time required for MS diagnosis and also reinforces the hypothesis that the virus might contribute to triggering the clinical onset of MS rather than starting the dysimmune reaction responsible for the “biological” onset of the disease itself.

All the described cases showed oligoclonal bands, except for three papers that did not report any data regarding CSF [53,54,57].

With respect to post-vaccinal MS cases, information on the type of vaccine and doses were available in all papers but for the case series from Lee et al. [51]. Forty-eight pwMS were administered mRNA-based vaccines, six pwMS were given a viral-vector vaccine and one had an inactivated virus-based vaccine. A total of 29 cases presented with neurological symptoms compatible with MS onset/reactivation after the first dose of vaccine, whereas in 18 cases disease onset/reactivation occurred after the second vaccination.

With respect to the clinical presentation of post-vaccinal cases, 4 had a supratentorial involvement, 12 presented with unilateral ON, 7 with symptoms related with an infratentorial involvement and 22 with symptoms suggestive of spinal cord dysfunction, 5 of which had multisymptomatic onset in which the spinal cord was always included. The simultaneous presence of CE and non-CE lesions is reported in all cases, as well as OB within the CSF, when specified. Moreover, the majority of cases presenting with a relapse after vaccine exposure were on DMTs when the symptoms occurred.

Both from our experience and a review of the cases reported in the literature, viral/vaccine exposure is more strongly associated with MS onset than with disease reactivation [49,51,52,53,54,55,56,57,58,59,60,61,62,63,64,65,66,67,68,69,70,71,72,73,74,75,76]. The most plausible explanation is the protective effect of DMTs against the inflammatory activation of the immune system known to be triggered by viral infection/vaccination. Indeed, among our cases, three out of four patients presenting with a clinical relapse were not on any kind of DMT.

Our cases as well as the published case reports share a temporal association between the exposure to SARS-CoV-2 or its related vaccines and MS onset/relapse, with a mean interval time between the former event and the latter of 2 weeks for our patients (1–55 days, the literature data). The temporal link between the two events is the only essential, although not sufficient, criterion defined by the WHO to establish a causal relationship between a dysimmune reaction and vaccination [77].

Cases of MS onset temporally following SARS-CoV-2 infection/vaccine are characterized by the subacute appearance of clinical symptoms, often associated with CE lesions in the MRI. However, the simultaneous presence of non-CE lesions and oligoclonal bands proves that the virus or the vaccine are not directly responsible for the dysimmune reaction but rather are related to the clinical unmasking of it. Whether the virus acts directly on the CNS or through abnormal peripheral immune activation is a matter of debate.

It is now ascertained that there are two possible CNS entry routes for SARS-CoV-2: hematogenous, through the disrupted blood–brain barrier (BBB) as well as through infected leukocytes that act as Trojan horses [78,79], or neuronal, through the olfactory nerves with subsequent neuron-to-neuron transmission through the olfactory bulb [80] or via the vagal nerve [81,82,83].

Once within the brain, SARS-CoV-2 spreads among different regions, including the thalamus and the spinal cord, through retrograde neuronal migration [84,85]. Several experimental studies have described an altered CNS environment during COVID-19, with a strong proinflammatory milieu leading to BBB disruption and the involvement of all the cellular components residing within the CNS, from astrocytes to oligodendrocytes, microglial cells and neurons [86,87].

Immune-mediated mechanisms such as molecular mimicry, epitope spreading and bystander activation of T lymphocytes, well-described phenomena usually implicated in autoimmunity development, have also been listed among the possible factors playing a role in the MS onset/relapse after COVID-19 [86,88].

Moreover, several potential immune-based intersecting pathways between SARS-CoV-2 infection and MS have been identified [17]: COVID-19 inhibits the production of IFN-1, a group of cytokines involved in the regulation of a T-cell-mediated immune response, which are also functionally altered in MS [89]. Indeed, IFNβ-1 was the first treatment approved for MS, and pwMS on IFN-1 had proven to be at a lower risk for COVID-19 [90]. Second, COVID-19 severity is associated with high levels of IFN-γ, IL-6 and IL-17, cytokines that are also involved in MS pathogenesis. Furthermore, both lung and enteric cells are infected with SARS-CoV-2, with an altered lung–gut axis that is postulated to be one of the potential pathogenetic mechanisms for MS. Lastly, the inflammasome NLRP3, a multiproteic complex residing in the cytosol of innate immune system cells, is hyperactivated in COVID-19, as well as in MS, where it promotes the release of proinflammatory cytokines responsible for both the increased virulence of SARS-CoV-2 and the chronic inflammation typically underlying tissue damage in MS.

Altogether, the cases described so far, including ours, do not have the epidemiological power to draw any firm conclusions on a causal relationship between the virus and MS. However, the temporal association between the viral infection and the onset of neurological symptoms, together with the experimental evidence of several common inflammatory pathways shared by the virus itself and MS, tends to advocate in favor of a role of SARS-CoV-2 as a trigger of MS clinical onset.

Altered activation of the immune system might also be implicated in the pathogenesis of MS cases reported to occur after SARS-CoV-2 vaccine exposure. The immune activation induced by vaccines is qualitatively and quantitatively different from the one evoked by the direct viral infection, as it is of lower intensity and skewed toward B cells. T cells, although activated by the vaccine, are less engaged and have the prominent function of supporting B-cells expansion, to ensure an antibody-mediated immunity [91].

Two main types of SARS-CoV-2 vaccines have been developed: viral-vector vaccines (i.e., AZD1222 from AstraZeneca and Ad26.COV2.S from Janssen) and mRNA-based vaccines (i.e., BNT162b2 by Pfizer BioNTech and mRNA -1273 by Moderna) [18,21,23,92]. According to Italian regulation, the former was recommended for fragile patients, including those with MS. In both cases, genetic information useful to encode the Spike protein of SARS-CoV-2 is introduced in human cells that start expressing the Spike antigen on the cell surface. This activates first the innate immune system response, boosted by the presence of different vaccine adjuvants, and later the adaptive immune response that becomes more prompt after the second dose of the vaccine. Interestingly, two of the cases here described presented only with mild sensory disturbances after the vaccine, whereas a subsequent COVID-19 infection was associated with severe clinical manifestations that led to the diagnosis. Moreover, one case of post-COVID-19 MS onset as well as the only COVID-19-related MS relapse had previously completed the recommended vaccination cycle, with no clinical manifestation of any sort. These findings reinforce the hypothesis of a less intense immune activation after vaccinal exposure, as previously hypothesized [93,94].

With respect to the clinical symptoms at onset, two of our post-COVID-19 cases manifested with optic neuritis (ON), two with an infratentorial impairment (one at the level of the brainstem and one cerebellar) and one presented with spinal cord dysfunction. Interestingly, all post-vaccine cases presented with sensory disturbances attributable to the spinal cord, associated with compatible MRI evidence of disease activity.

Intriguingly, the pattern of clinical manifestations of all the published case reports very frequently consisted of optic neuritis and brainstem and spinal cord involvement (mostly sensory pathways), in agreement with our observation. Whereas ON is one of the most common clinical manifestations at MS onset, the proportion of cases with brainstem or spinal cord impairment both after COVID-19 (Table 1) and after vaccine exposure (Table 2) seems unusually high.

**Table 2 viruses-15-01569-t002:** Clinical and MRI features of demyelinating disorders onset/relapse after vaccine exposure.

Authors	Vaccine Type/Dose	Time Interval	Symptoms	CE Lesions	Non-CE Lesions	OB	DMTs	§§ Anti-MOG/AQP4 Titration
**MS ONSET**								
Case 6	BNT/2nd	2 weeks	Multisymptomatic (gait/trunk ataxia, sphincter)	(MRI not in acute phase)	+	+		-/-
Case 7	ChAdOx1/1st	1 week	SC	+	-	+		-/-
Case 11	BNT/1st	1 week	SC	SC	+	+		NA
Case 12	BNT/1st	2 weeks	SC	SC	+	+		+/+
Lee et al. [51] *	Various	0–55 days	Mainly SC	Various sites	variable	various		+/+
Czarnowksa et al. [62]	JJ/1st	2 weeks	SC	Brain, SC	+	+		+/+
Havla et al. [95]	BNT/1st	1 week	SC	SC	+	+		+/+
Khayat-Khoei et al. [64] 3 cases	mRNA vaccines/ 2nd (2)	2–3 weeks	ON (2) SC (1)	Brain (1) SC (2)	+	NA		-/+
Kaulen et al. [65]	BNT (4)	2 weeks	ON (4) SC (2) ST (2)	ON (4) SC (2) ST (2)		+		+/+
Mele et al. [66]	Mod/1st	2 days	Cerebellum	ST	+	+		−/+
Rinaldi et al. [68] 2 cases	BNT/1st (1) 2nd (1)	1 week	BS+SC (1) ST (1)	BS+SC (1) ST+ SC non-ce (1)	+	+		+/+
Gernert et al. [63] 5 cases	BNT (4, 3 cases 1st, 1 case 2nd), 1 ChadoOx/2nd	1–2 weeks	ON (2) BS (1) SC (1) BS+SC (1)	SC (2) BS/cerebellum (3)		+		+/+
$ Watad et al. [69] 4 cases	BNT/1st (2) Mod/2nd (2)	1 day–1 month	ST (1); cerebellum, SC (1) SC (2)	+	+	+		+/+
$ Nistri et al. [67] 2 cases	ChadOx/1st	1 week	ON (2) IT (1)	+ supratentorial	+	+		-/-
Toljan et al. [70] 4 cases	BNT/1st (2)		Uncertain	+	+	+		-/-
**MS RELAPSE**								
Rinaldi et al. [68]	BNT/1st	2 weeks	Cerebellum/BS	ST	+	+	IFNb	NA
$ Al-Midfai et al. [71]	JJ/2nd	2 weeks	Uncertain (unilateral motor deficit)	-	+	NA	-	NA
$ Maniscalco et al. [72]	BNT/1st	2 days	Unilateral SM deficit	ST	+	NA	FTY	NA
Kataria et al. [73]	BNT/2nd	3 weeks	ON + SC	ST	+	-	IFNß	NA
Seyed Ahadi et al. [74]	Sinopharm/1st	2 days	Paraparesis and ataxia	BS	+	NA	-	NA
Kayat-Khoei et al. [64] 4 cases	mRNA vaccines 1st (1) and 2nd (3)	1–3 weeks	ON (2) BS+SC (1) Possibly SC (1)	Brain (3)	+ (new SC in 2 cases)	NA	NTZ (1) FTY (1) None (2)	NA
$ Nistri et al. [67] 14 cases	ChadOx/1st (2) mRNA/1st (7), 2nd (5)	2 days–3 weeks	BS/cerebellum (4) SC (6) Possibly SC (6)	Brain (13)	+	+	+ (9) - (5)	NA
D’apolito et al. [75]	BNT/2nd	4 days	BS/cerebellum	+	-	+	FTY	NA
$ Fragoso et al. [49] 8 cases	BNT/1st	BS	Various	Some	NA	NA	Various DMT (7)	NA

Legend: BS: brainstem; ON: optic nerve; NA: not available; ST: supratentorial; SC: spinal cord. * case series; $: spine MRI data not available; §§ in order to be included in this paper, all the cases had to result negative to anti-MOG or antiAQP4 antibodies. Therefore, the results in the table refer to +: titled or -: not titled.

Along with MS cases that occurred after vaccine exposure, some authors also reported post-vaccinal cases of transverse myelitis together with cases of neuromyelitis optica presenting with simultaneous brainstem and spinal cord impairment [68]. Furthermore, transverse myelitis is the most common clinical manifestation in case series of post-vaccinal CNS demyelinating disorders, and some cases of isolated myelopathies without brain involvement have been described [68,96,97,98].

Interestingly enough, a neuropathological study on the post-mortem brain of SARS-CoV-2-positive subjects revealed the presence of activated microglia and cytotoxic T lymphocytes mostly in the brainstem and cerebellum, together with SARS-CoV-2 viral proteins in cranial nerves and isolated cells originating from the lower brainstem [81]. This would be an argument for direct viral-mediated damage and also would plausibly explain the high occurrence of brainstem symptoms at onset.

In some cases of post-COVID-19 myelopathies, an extensive electrophysiological study revealed grey matter spinal damage, more likely compatible with direct viral detrimental action within the spinal cord [96]. Furthermore, experimental evidence of a murine coronavirus, entering the CNS through the olfactory bulb and rapidly spreading to the spinal cord through astrocyte-to-astrocyte transmigration, indirectly supports the possibility of viral-mediated spine tissue damage after COVID-19 [84].

Alternative hypotheses range from the possibility of molecular mimicry between viral antigens/adjuvants contained in vaccines and specific cell-surface receptors located in the spine, to the activation of a proinflammatory cascade involving immune system components that preferentially locate/act in specific CNS areas, such as the brainstem and spinal cord.

These findings need to be confirmed in larger studies with a better characterization of the cellular and humoral immunological factors activated by both the virus and related vaccines.

## 8. Conclusions

In conclusion, we here present a case series of CNS inflammatory disorders occurring in temporal association with SARS-CoV-2 infection and the related vaccines. Viral or vaccine exposure is more frequently associated with the clinical onset of MS rather than with MS reactivation, likely due to the protective effect of disease-modifying treatments in clinically definite MS cases. The anatomical CNS districts involved in the clinical manifestations of all the cases published as of now, including ours, are mainly represented by optic nerves, the brainstem and the spinal cord. Further studies investigating the possible role of genetic factors and specific immune pathways in the occurrence of dysimmune CNS disorders after viral/vaccines exposure, as well as explaining the tropism for specific CNS regions are needed.

## Figures and Tables

**Figure 1 viruses-15-01569-f001:**
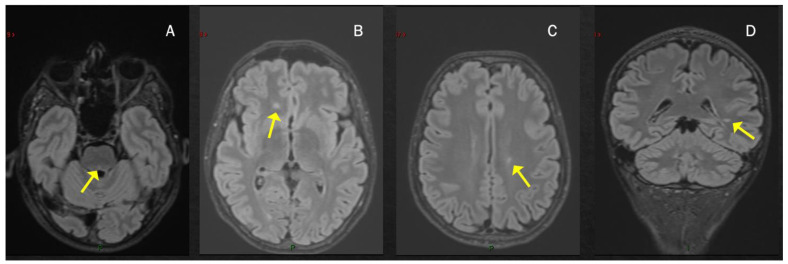
T2-hyperintense lesions located infratentorially in the dorsal portion of the pons, see arrows (**A**) and in the supratentorial region with a periventricular distribution (**B**–**D**).

**Figure 2 viruses-15-01569-f002:**
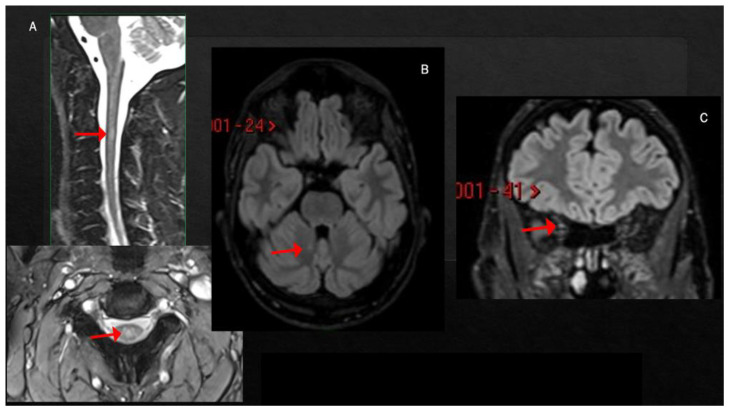
T2-hyperintense lesion at the level of C2–C3, involving the central portion of the spinal cord (**A**). Single small lesion in the paravermis of the left cerebellum (**B**) and mild T2-hyperintensity of the right optic nerve (**C**).

**Figure 3 viruses-15-01569-f003:**
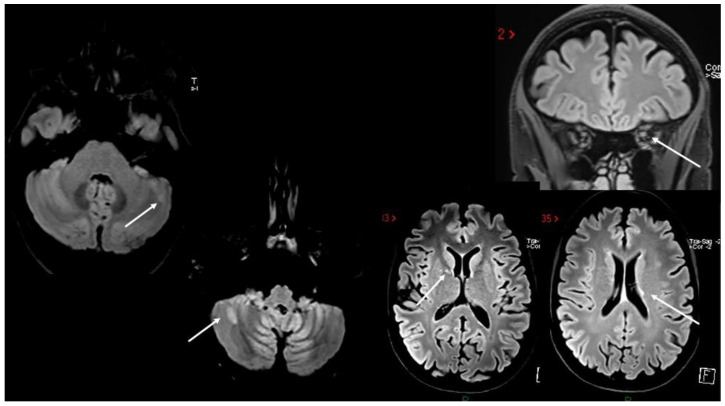
White matter lesions in both supratentorial and infratentorial regions, with concomitant swelling and T2-hyperintensity of the left optic nerve.

**Figure 4 viruses-15-01569-f004:**
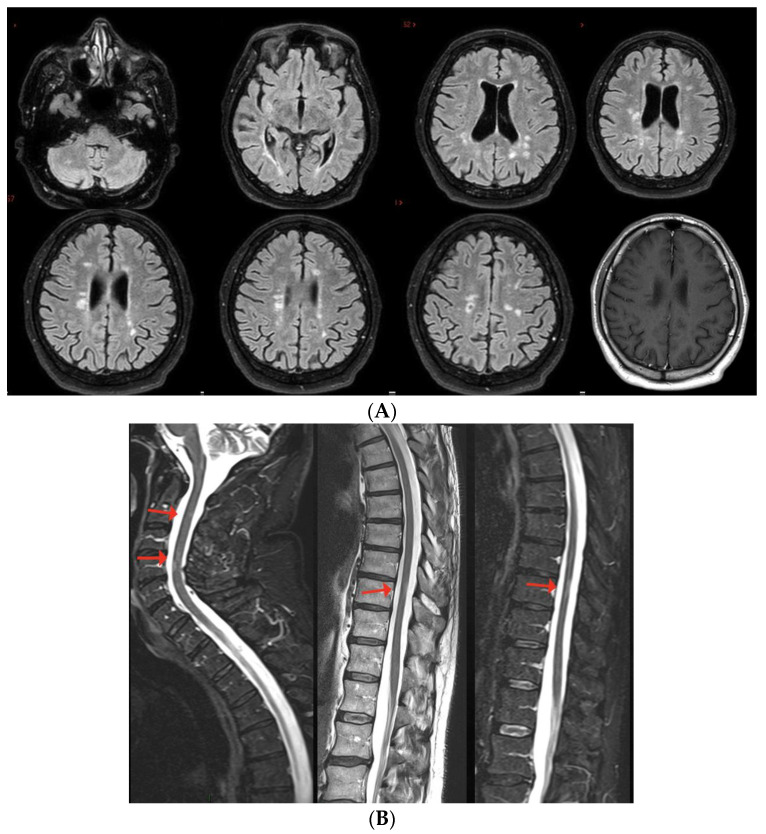
(**A**,**B**) Multiple non-enhancing lesions in both brain and spinal cord.

**Figure 5 viruses-15-01569-f005:**
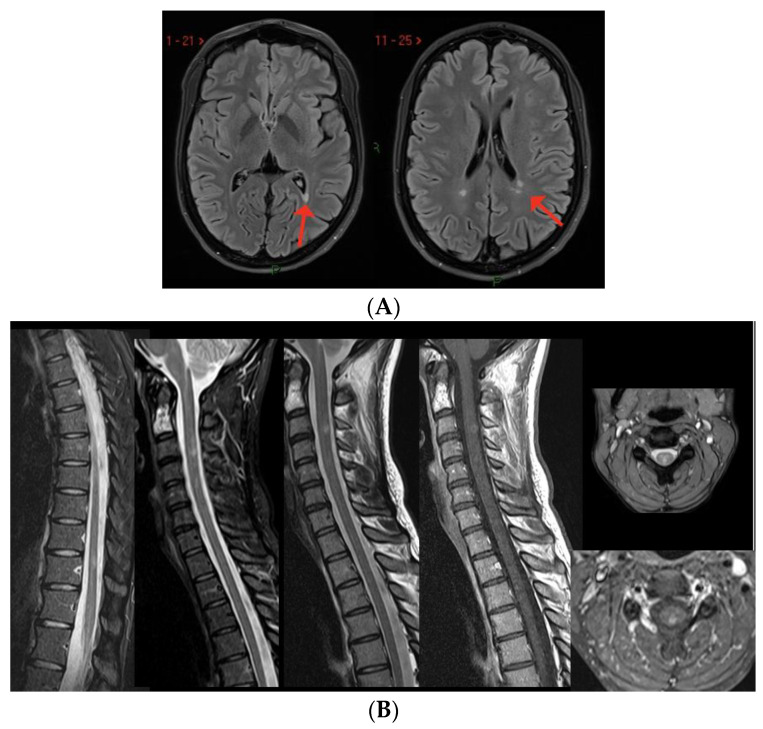
(**A**) Three new non-enhancing lesions in the periventricular white matter. (**B**) Spine MRI showing one active lesion in the cervical spinal cord and two non-enhancing dorsal lesions.

**Figure 6 viruses-15-01569-f006:**
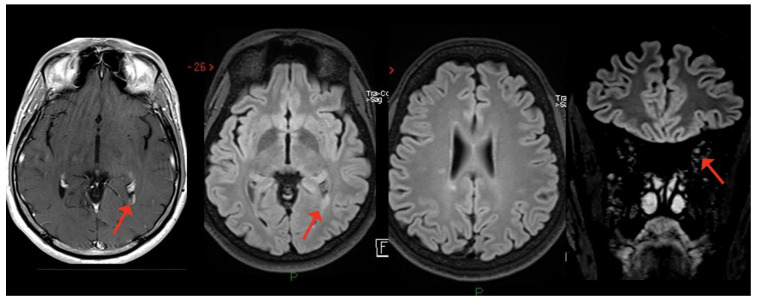
Multiple white matter lesions in the brain, with gadolinium enhancement of one lesion in the periventricular region; slight T2-hyperintensity with no enhancement of the left optic nerve.

**Figure 7 viruses-15-01569-f007:**
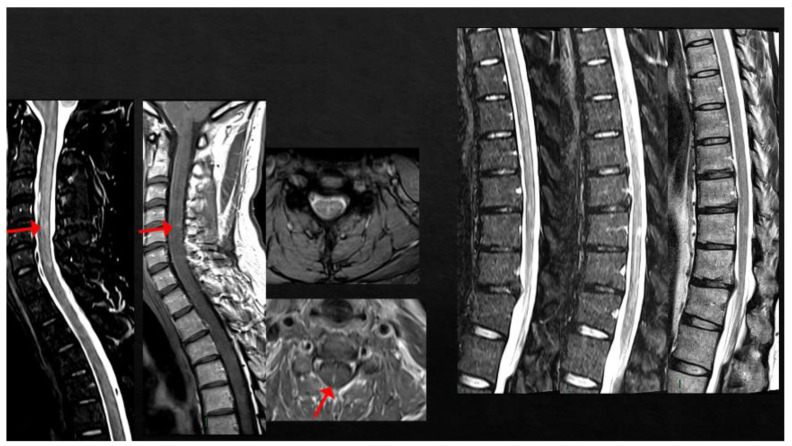
Multiple cervical and thoracic spinal cord lesions, with contrast enhancement at the level of C2 and C4-C5.

**Figure 8 viruses-15-01569-f008:**
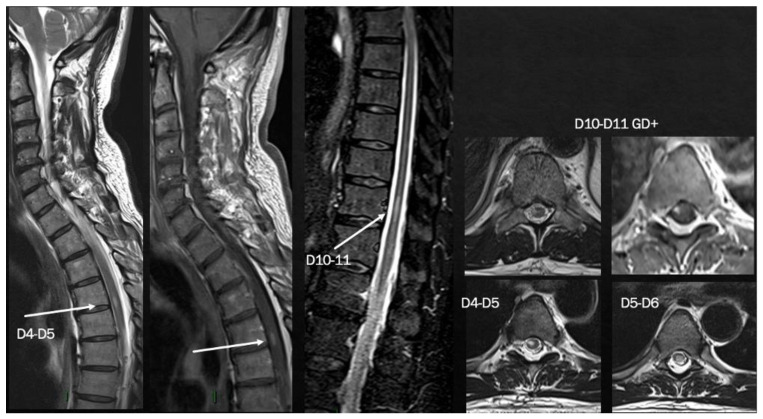
Brain MRI shows multiple T2-hyperintense lesions in both supra- and infratentorial regions. Spinal cord MRI documents a predominantly dorsal lesion burden. After gadolinium injection, at least 7–8 (many) lesions show peripheral or homogeneous enhancement in both brain and spinal cord.

**Table 1 viruses-15-01569-t001:** Clinical and MRI features of post-COVID-19 MS onset/relapse.

Authors	Time Interval	Anatomical Location Related to Clinical Symptoms	CE Lesions	Non-CE Lesions	OB	Previous Vaccine	§§ Anti-MOG/AQP4 Titration
**MS ONSET**							
Case 1	2 weeks	BS (dyplopia)	-	+	+	-	+/+
Moore et al. [52]	2 weeks	BS (dyplopia)	BS, ST	+	+	NA	+/+
Yavari et al. [53]	3 weeks	BS (dyplopia)	ST	+	NA	NA	-/-
Carta et al. [54]	concomitant	BS (dyplopia)	ST ^	+	NA		-/-
Pignolo et al. [55] 1st case	8 weeks	BS (facial palsy)	-	+	+	-	+/+
Ismail et al. [76].	8 weeks	Cerebellum	-	+	+	-	-/-
Case 2	2 weeks	ON	- SC	+	-	-	+/+
$ Case 3	2 weeks	ON	-	+	+	-	+/+
Palao et al. [56]	2 weeks	ON	ST	+	+	-	+/+
Sarwar et al. [57]	3 weeks	ON and unilateral motor deficit	ST	+	NA	NA	-/-
Case 4	4 weeks	SC	SC	+	+	+	+/+
Fragoso et al. [58]	6 months	SC	ST, SC	+	+	-	+/+
Avila et al. [59] *	2–6 weeks	40% SC 40% ON 20% BS	NA	NA	+	80% - 20% +	NA/+
Feizi et al. [60]	1 week	SC	SC	+	+	NA	+/+
**MS RELAPSE**							
Pignolo et al. [55] 2nd case	2 months	Uncertain (unilateral motor symptoms)	ST	+	+	-	NA
$ Case 5	3 weeks	Cerebellum (trunk/gait ataxia, dysmetria)	NA	NA	NA	+	NA
Finsterer [61]	2 weeks	SC (sensory symptoms both inferior limbs and trunk level)	SC	+	+	-	NA

Legend: BS: brainstem; ON: optic nerve; NA: not available; ST: supratentorial; SC: spinal cord. * Case series; $: spine MRI data not available; ^ a lesion located in the pons compatible with symptoms was found but did not show contrast enhancement; §§ in order to be included in this paper, all the cases had to test negative to anti-MOG or antiAQP4 antibodies. Therefore, the results in the table refer to +: titled or -: not titled.

## Data Availability

The data presented in this study are openly available in Zenodo at 10.5281/zenodo.8143310.

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
