# Peer review of "The Potential Role of SARS-CoV-2 Infection and Vaccines in Multiple Sclerosis Onset and Reactivation: A Case Series and Literature Review"

_viruses, 2023, doi:10.3390/v15071569_

Round 1
Reviewer 1 Report
I thank the authors for a very interesting review. They described a really very important problem. The title of the article states “a case series and literature review”, but the article does not provide a review of the described cases of the relationship between SARS-CoV-2 and MS. More attention should be devoted to this issue.
In addition, there are a number of small comments, which are listed below:
Introduction
1. Please put a period after the link. Not: “.[8-10]”, right: “[8-10]”.
2. “In particular, with respect to multiple clerosis (MS), a recent study reported a close relationship between Epstein-Barr virus (EBV) infection and MS prevalence in a large sample of U.S. veterans.” You describe only one study that showed an association between MS and EBV, but this fact has been reported by other researchers as well. Please provide links to other studies.
3. After [11]: Please add literature data on the identified cases of the relationship between SARS-CoV-2 and MS.
4. "In the MS field". This is not a good phrase, please reformulate.
5. What are “anti-CD20 therapies”? Why is it important to compare the group receiving this therapy and not receiving it? Are there studies for groups of MS patients receiving other treatments?
6. Case 1: “CSF examination showed 15 lymphocytes…” What do you mean by 15 lymphocytes?
7. Case 3: Please provide information on oligoclonal bands.
8. Case 5: Please make a conclusion about the patient.
9. Cases 1-5: How is COVID-19 detected?
10. Case 10: Was MS diagnosed? It should be described.
Discussion
11. Please decipher “DMTs”
12. “...criterion defined by WHO to establish a causal relationship between immunization and vaccination.” Vaccination is also immunization. Replace “immunization” with convalescents or recovered COVID-19
13. “Cases of MS onset temporally following SARS-CoV-2 infection/vaccine are characterized by the subacute appearance of clinical symptoms, often associated with CE-lesions at MRI but also showing non-CE lesions and oligoclonal bands, proving that the virus or the vaccine are not directly responsible for the dysimmune reaction but are potentially related to the clinical unmasking of it.” This proposal is not clear. Please reformulate.
14. A conclusion section should be added.
Author Response
Dear Editor,
we hope we were able to address all the queries from Reviewer 1.
We thank the Reviewer for the opportunity to revise and improve the manuscript. As suggested by the Reviewer, we added a whole section in the Discussion providing information on all the published cases of MS onset/reactivation after viral/vaccine exposure. We hope we were able to correctly address all the issues.
- Please put a period after the link. Not: “.[8-10]”, right: “[8-10]”.
R: Changes have been made accordingly
- “In particular, with respect to multiple clerosis (MS), a recent study reported a close relationship between Epstein-Barr virus (EBV) infection and MS prevalence in a large sample of U.S. veterans.” You describe only one study that showed an association between MS and EBV, but this fact has been reported by other researchers as well. Please provide links to other studies.
R: Some other references on the same topic have been added
- After [11]: Please add literature data on the identified cases of the relationship between SARS-CoV-2 and MS.
R: some literature references have been added here together with a sentence underlining the potential role of SARS-CoV-2 as a trigger of neuroinflammation
- "In the MS field". This is not a good phrase, please reformulate.
R: the phrase “in the MS field” has been substituted with “In the context of MS”
- What are “anti-CD20 therapies”? Why is it important to compare the group receiving this therapy and not receiving it? Are there studies for groups of MS patients receiving other treatments?
R: We thank the Reviewer for the comment. Anti-CD20 therapies, as now specified in the text, are monoclonal antibodies specifically targeting B lymphocytes, the immune cells responsible for antibodies production, among other functions. These patients have been carefully observed during the pandemic as they were though to be at a higher risk of developing severe COVID-19 infections, as well as at risk of not developing an effective immune response after vaccinations.
- Case 1:“CSF examination showed 15 lymphocytes…” What do you mean by 15 lymphocytes?
R: we specified that the CSF revealed the presence of 15 cells, that were only lymphocytes
- Case 3: Please provide information on oligoclonal bands.
R: We are sorry to not understand the question. It is already stated that CSF analysis showed the presence of oligoclonal bands, like in the other cases.
- Case 5: Please make a conclusion about the patient.
R: We added that the patient has been clinically stable since then
- Cases 1-5: How is COVID-19 detected?
R: we specified that all these cases were diagnosed through a nasopharyngeal swab for SARS-CoV-2 reverse-transcriptase-polymerase-chain-reaction assay
- Case 10: Was MS diagnosed? It should be described.
R: we thank the reviewer for the question. It was already specified in the abstract but we added it also in the final section of the introduction, that this case series is about 11 MS cases and 1 case of myelitis, that is, indeed, case 10
Discussion
- Please decipher “DMTs”
R: we spelled out MTs in case 3, the first time it appears in the text and we thank the Reviewer for the comment
- “...criterion defined by WHO to establish a causal relationship between immunization and vaccination.” Vaccination is also immunization. Replace “immunization” with convalescents or recovered COVID-19
R: We thank the Reviewer for the comment, allowing us to make this sentence clearer. We meant to state that the temporal association between vaccination and the dysimmune disorder occurring subsequently is the only essential criterion. If the reviewer agrees, we would substitute the word “immunization” with “dysimmune reacrion”
- “Cases of MS onset temporally following SARS-CoV-2 infection/vaccine are characterized by the subacute appearance of clinical symptoms, often associated with CE-lesions at MRI but also showing non-CE lesions and oligoclonal bands, proving that the virus or the vaccine are not directly responsible for the dysimmune reaction but are potentially related to the clinical unmasking of it.” This proposal is not clear. Please reformulate.
R: We reformulated the sentence as follows: “Cases of MS onset temporally following SARS-CoV-2 infection/vaccine are characterized by the subacute appearance of clinical symptoms, often associated with CE-lesions at MRI. However, the simultaneous presence non-CE lesions and oligoclonal bands prove that the virus or the vaccine are not directly responsible for the dysimmune reaction but rather, are related to the clinical unmasking of it”
- A conclusion section should be added.
R: we separated the conclusion in a specific section and added some conclusive sentences.
Reviewer 2 Report
The focus of this review is to discuss the potential association between viral/vaccine SARS-CoV-2 exposure and MS onset/relapses, through data from the literature and 12 clinical cases. An important message is that SARS-CoV-2/vaccine exposure is more strongly associated with MS onset than with disease reactivation, which is consistent with literature data. The authors reported also an unusually high frequency of brainstem and spinal cord impairments in MS patients after SARS-CoV-2 infection of after vaccine administration.
The sections of the manuscript are well organized, and the tables presented are relevant. The authors have provided a nice and updated dissection of this intriguing potential association between SARS-CoV-2 (virus and vaccine) and MS onset.
Author Response
We thank the Reviewer for the comments
Round 2
Reviewer 1 Report
The new additions to the manuscript made a big difference. The quality of the paper had improved, and all my questions were addressed. No more comments.